# Systematic Investigations of Annealing and Functionalization of Carbon Nanotube Yarns

**DOI:** 10.3390/molecules25051144

**Published:** 2020-03-04

**Authors:** Maik Scholz, Yasuhiko Hayashi, Victoria Eckert, Vyacheslav Khavrus, Albrecht Leonhardt, Bernd Büchner, Michael Mertig, Silke Hampel

**Affiliations:** 1Leibniz Institute for Solid State and Material Research Dresden, Helmholtzstr. 20, 01069 Dresden, Germany; maik_scholz@web.de (M.S.); victoria-eckert@web.de (V.E.); vhavrus@gmail.com (V.K.); leoalb@web.de (A.L.); B.Buechner@ifw-dresden.de (B.B.); 2Institute for Physical Chemistry, Technische Universität Dresden, 01062 Dresden, Germany; michael.mertig@tu-dresden.de; 3Graduate School of Natural Science and Technology, Okayama University, 3-1-1 Tsushima-naka, Kita, Okayama 700-8530, Japan; 4Institute for Solid State Physics, Technische Universität Dresden, 01062 Dresden, Germany; 5Kurt-Schwabe-Institut für Mess- und Sensortechnik e.V. Meinsberg, 04736 Waldheim, Germany

**Keywords:** carbon nanotube yarns, carbon nanotube, functionalization, electrical conductivity, annealing, acid treatment

## Abstract

Carbon nanotube yarns (CNY) are a novel carbonaceous material and have received a great deal of interest since the beginning of the 21st century. CNY are of particular interest due to their useful heat conducting, electrical conducting, and mechanical properties. The electrical conductivity of carbon nanotube yarns can also be influenced by functionalization and annealing. A systematical study of this post synthetic treatment will assist in understanding what factors influences the conductivity of these materials. In this investigation, it is shown that the electrical conductivity can be increased by a factor of 2 and 5.5 through functionalization with acids and high temperature annealing respectively. The scale of the enhancement is dependent on the reducing of intertube space in case of functionalization. For annealing, not only is the highly graphitic structure of the carbon nanotubes (CNT) important, but it is also shown to influence the residual amorphous carbon in the structure. The promising results of this study can help to utilize CNY as a replacement for common materials in the field of electrical wiring.

## 1. Introduction

To fulfill future claims on everyday applications for higher efficiency, new materials with improved physical properties and lower production costs are necessary. One of the most promising candidates for these materials is carbon nanotubes (CNT). These one-dimensional tubular carbon structures have been shown to possess an impressive array of physical properties on the scale of individual tubes [1], including but not limited to ballistic electron transport [2], high thermal conductivity [3] and mechanical strength [4]. Together with their low density and high current carrying capacity [5], they outperform most commonly used materials such as copper. Translating these outstanding properties from the nano to the bulk scale is therefore understandably a major focus in the field of CNT research. Besides CNT hybrid systems with polymers, spinning macroscopic yarns out of different CNT starting materials like sheets and arrays is a promising way to incorporate these properties into practical materials. These carbon nanotube yarns (CNY) have a high tensile strength [6], are extremely flexible and very light weight, which makes these materials promising candidates for electrical wiring in different applications [7,8,9,10,11]. However the physical properties, especially the electrical conductivity, still lag behind individual CNT and copper [1].

The main factors defining the electrical conductivity of CNY are the intrinsic electrical properties of the CNT and the contact resistant between adjacent CNT. Different groups have shown that adjusting the CNT type used for spinning can improve the electrical properties of these yarns. An ideal CNY would consist purely out of long metallic single walled CNT and reach conductivity as high as an individual CNT [12]. Other ways to enhance the electrical properties of CNY include post-spinning treatments such as annealing and chemical functionalization. Annealing is known to repair structural defects in CNT [13,14] and is a reagent free way to improve the electrical properties of CNY [15,16]. Different approaches to anneal CNY have been developed over the years [15,17,18]. However, because CNT are only connected through weak van der Waals forces where functionalization by an acid treatment can only influence the inter-tube interactions by means of surface modification. It has been claimed that introducing oxygen rich groups raise the charge carrier density between adjacent CNT [19,20]. There is an ongoing controversy if functional groups are the main factor for increasing electrical conductivity or the change of yarn diameter through this acid treatment [21,22].

In this work, we provide a systematical study on the effects of annealing and functionalization by acid treatment to CNT yarns. Hereby different acids were tested for functionalization as well as annealing temperatures up to 2500 °C.

## 2. Results and Discussion

### 2.1. Annealing

The pure multiwalled CNT (MWCNT) array was shown to consist of CNT with 2 to 6 walls with the majority of CNT possessing 2 to 4 walls [23]. The diameter of pristine yarn was around 22 ± 1 µm (Figure 1 and Appendix A).

For improvement of the properties of CNY and therefore the electrical conductivity the CNY were annealed under an Argon atmosphere at 1000, 1500, 2000 and 2500 °C for 30, 60 and 120 min (Figure 2). The appearance of all CNY after annealing differs from the untreated yarn with a higher density surface, lower spacing and fewer voids between the CNT bundles (Figure 2c). Additionally, the diameter decreased about 1 µm. After annealing at 2500 °C CNT with 4 to 6 walls make up the majority—over 70 % of the observable CNT (Figure 2a and Appendix A). The increase in the number of walls is a process that has also been observed in other works [24]. This behavior can be explained by the increased thermal activity of the outer wall and the desire of the system to take a state of energetic minimum. When examined with TEM, annealed CNT show a much straighter structure compared to that of the pristine yarn (Figure 2d). There are also fewer recognizable defects in the wall structure. This is consistent with reports that high-temperature treatment mainly influences the microstructure of the CNT or CNT walls [25].

The improvement in CNY microstructure is also supported by Raman measurements. With increasing time and temperature the I_D_/I_G_ ratio drops from 0.83 to 0.23 for samples annealed at 2500 °C for 2 h (Figure 3 and Figure 4).

Increased graphitization becomes noticeable after 1 h annealing at 1500 °C with a decrease of the I_D_/I_G_ ratio to 0.68. After 2 h of annealing at 1500 °C the ratio drops to 0.45, which corresponds to almost 50% of the untreated yarn sample. This shows that it is not only temperature, but also the time of annealing that influences the increasing the graphitization of this material (Figure 3 and Figure 4).

Improved crystallinity via high temperature annealing is also in good agreement with previously reported results for CNT yarns [15,17] and individual CNT [13,14]. Comparing the results of this study to previous work [17], it is suggested that other annealing techniques such as resistance-heating in vacuum might improve the crystallinity even more. This may be linked to the fact that the conventional annealing used for this work does not result in the observable evaporation of amorphous carbon from the sample. This fact may play an important role in the quality of the treated samples, as we will discuss below in the results of electrical conductivity.

Looking at the behavior of the electrical conductivity, it is noticeable that in the same ratio as the I_D_/I_G_ ratio is decreasing the conductivity at room temperature increases (Figure 4a,b). Electrical conductivity is enhanced from 300 S/cm for pristine yarn to ~1680 S/cm for the yarn sample annealed for 2 h at 2500 °C. This is a significant increase of ~460% in total for a single treatment.

Because of the higher crystallinity of the sample, it is thought that a lack of defects in the CNT walls and therefore less scattering points for electrons within the individual CNT might be related to this increase in conductivity. But another aspect is here also relevant. Comparing these results to our previous work, we still find amorphous carbon within the CNT after annealing. This carbon will function as a conductive bridging agent between adjacent highly graphitic CNT (Figure 5) and provide a pathway for conduction of electrons. According to the so-called Mott variable range hopping (VRH) model, the electrons jump from one starting point to another with the lowest possible hopping energy. The electrons must overcome the gap between one CNT and the other. Amorphous carbon is therefore helping to bridge this process. The whole process a CNY is an interplay between a high crystalline and high conductive CNT and the next CNT. Here a compressed CNY structure (like after annealing or acid functionalisation) as well as additional carbon as a conductive bridge is ideal to overcome the gaps by hopping [26].

Figure 6a,b show specific and normalized electrical conductivity of CNT yarns annealed for 2 h at different temperatures in the range of 5 to 295 K. Each sample shows monotonically increasing conductivity with temperature, which is typical for MWCNT materials [27,28,29]. The electrical conductivity also increases over the whole temperature range with increasing annealing temperature. In comparison to Kaiser et al. [30] and Skákalová et al. [31] the curves of normalized conductivity indicate that the predominant conduction mechanism for both the pristine and annealed CNY is three-dimensional variable range hopping. This mechanism describes a phonon-assisted tunneling process between localized charge carrier states [32,33]. It is, therefore, thought that the hopping takes place between occupied and unoccupied states that are separated both spatially and energetically from each other. As the temperature of the sample decreases, the thermal energy of the phonons recede and fewer states become energetically attainable, thus the electrical conductivity drops. With increasing annealing temperature the normalized temperature dependent conductivity measurements show a slight decrease of the slope with increasing annealing temperature (Figure 6b).

### 2.2. Acid Treatment

To further increase the electrical conductivity the CNY were functionalized. A variety of different acids were used and the acid treatment was carried out at room temperature for 3 to 216 h for each of the acid treatment processes.

Oxidative acids like HNO_3_ have been used to purify CNT and introduce functional groups on their surface. Diluted nitric acid has been used to purify CNT [34] whereas concentrated or mixtures of strong acids effectively introduce oxygen rich groups onto CNT [35]. The effects of acidification time on the performance of CNT yarns was examined, with different high concentrated acids, namely H_2_O_2_ (30%), HCl (37%), HNO_3_ (65%) and half concentrated HNO_3_ (32%).

After acid treatment, the appearance of the yarn differs strongly depending on the kind of acid and treatment time. The least difference in appearance between the pristine yarn and the acid treated were observed after the H_2_O_2_ treatment. Even after 216 h, the yarn structure does not differ significantly from the original sample (Appendix A). Due to capillary forces during drying of the yarns after treatment no loose CNT bundles are noticeable. After 216 h HCl treatment leads to a deformation of the yarn in the direction of twisting. Short treatment times, however, do not lead to any significant change in the appearance (Appendix A). In the case of treatment with 50% concentrated HNO_3_, the CNY shows clear deformations and indentations along the direction of twist after a short treatment time of 3 h. Increasing treatment time furthers the deformation of the yarn (Figure 7). Nevertheless, the surface remains closed and without voids or gaps (Figure 7b,d). As with H_2_O_2_ and HCl, the capillary forces occurring during drying after acid treatment are the cause. It should also be noted that the twist of the yarn is maintained under these conditions even after a treatment time of 216 h.

Treatment with concentrated HNO_3_ results in a dramatic change of appearance. Even after a short treatment time, significant indentations and deformations of the yarn are observable. The twist of the yarn is already no longer recognizable after 3 h treatment. However, the surface of the yarn remains dense and closed as with the treatments by the other acids. No major gaps or similar defects appear. As the treatment time increases, the deformation of the yarn increases and in addition to the depressions along the fiber direction, the yarn also vertically folds perpendicular to the fiber direction occur (Figure 8). However, after a treatment time of 216 h some, but not all, of the outer layer of the yarn appears to have broken off (Figure 8c inset). This indicated that the outer layer is no longer as tightly bound at the remaining inner part of the yarn but only a quarter of the examined length shows this behavior. The other sections of the yarn surface appear to remain closed, as with treatment with the other acids (Figure 8d).

These results show that CNT yarns are attacked and deformed differently by different acid treatments. There is a ranking of H_2_O_2_, HCl, half and concentrated HNO_3_, where concentrated HNO_3_ causes the largest morphology changes. One cause of these deformations is thought to be insufficient compaction of the yarn during spinning. This may leave voids inside the yarn structure, which could be are compressed by the surface tension of the acids or contracted by the capillary forces during the drying of the yarns.

In addition to the effect of acid treatment on the surface and the appearance of the yarn, the effects on the structure, diameter and electrical conductivity of the yarn were investigated. Figure 9 shows the results of these investigations for the different acid treatments as a function of time. One striking feature common to all the acids treatments is that the maximum increase of conductivity is achieved with a treatment time of 3 h. With increasing treatment time the conductivity drops to lower values and in the case of concentrated HNO_3_ even to the value of the untreated yarn (300 S/cm).

The behavior of the yarn after treatment with concentrated HNO_3_ is different from the other tested acids. There is a strong variation of the I_D_/I_G_ ratio but only a relatively small fluctuation of the average yarn diameter observed with this treatment. However, it should be noted that the measured diameter of the yarn is subject to high fluctuations, as indicated by the large error bars. These variations are due to the severe deformation and partial detachment of parts of the outer yarn layer. Over the considered experimental period (3 to 216 h), the I_D_/I_G_ ratio is subject to large variations ranging from 0.68 to 0.88. This, in turn, shows the strong impact of concentrated HNO_3_ on the structure of CNT materials. Electrical conductivity drops after the strong increase at 3 h and drops to 300 S/cm after 216 h. The maximum value for the conductivity is 642 S/cm, which represents an increase of more than 110%. For verification, a trial was carried out with only one hour of treatment, which only resulted in a value of 416 S/cm. With this, it appears that 3 h represents an optimal treatment time with concentrated HNO_3_ for the yarn used in this work. The drop of conductivity after 3 h treatment could be explained by the strong oxidative nature of concentrated HNO_3_, this results in a strong defect introducing behavior. This appears to be proof that the action of introducing functional groups on the surface of CNY is not as effective as claimed as increasing treatment time should result in a noticeable positive effect on conductivity by increasing the charge carrier density between the CNT. In addition to this, a stable densification of the yarn through the functional groups and introduced stronger dipol-dipol and H-bridge bonds should be observable [20].

The influence of the other acids (H_2_O_2_, HCl and half concentrated HNO_3_) is less dramatic than with concentrated HNO_3_, with the I_D_/I_G_ ratio being only minimally affected. It is even decreased by the influence of HCl. After 216 h, the values are in the range of 0.72 to 0.78. This decrease is explained in literature by the removal of carbonaceous impurities during acid treatment [36,37]. However, the nature of these carbonaceous impurities was not explained. It has been observed in this work that the yarn diameter and the electrical conductivity appears to show a strong correlation. This is best seen in the half concentrated HNO_3_ treated sample with the electrical conductivity increases with decreasing diameter and vice versa. The maximum values reached after 3 h are for H_2_O_2_, HCl and half concentrated HNO_3_ 587, 551 and 541 S/cm, respectively. In the case of H_2_O_2_ a rise in electrical conductivity (518 S/cm) can be observed after a treatment time of 216 h. However this effect has not yet been explained and requires further investigation. It might be assumed that temperature fluctuations and the influence of light during the experiment leads to a decomposition of H_2_O_2_ and formation of hydroxyl radicals that can promote the formation of functional groups on the surface of the CNT.

Different groups describe a similar positive effect on the electrical conductivity of CNT yarns by acid treatment [20,21,22,36,38]. However, the explanation for the increase in conductivity is different in each case. One of the most common explanations is that the influence of oxidative acids such as nitric acid or mixtures of nitric acid and other acids functionalizes the surface of the CNT with oxygen-containing groups such as hydroxyl, carboxyl and carbonyl groups. This increases the electron density between the CNT and creates additional conduction paths between the CNT [20]. Another explanation is that the smaller distance between the CNT after acid treatment reduces the contact resistance, by which the electron-hopping mechanism is supported [22]. Our experiments support the hypothesis of Meng et al. [22] where the formation of functional groups plays a secondary role compared to the compaction of the yarns. However, Meng et al. have described this effect only for concentrated HNO_3_. In our experiments, we show that other acids lead to a similar effect. Because of the hydrophobic nature of the CNT yarns, water-based solutions like the used acids can only minimally infiltrate the inner structure of the yarns when treated for a short time. It is therefore thought that the yarns get compressed from the outside by the surface tension of the solutions. This results in the inter-tube spacing becoming smaller, thereby reducing the contact resistance between CNT. A less significant effect was observed with water. Here the shrinkage of diameter and rise of conductivity isn’t as high as with the acids. This could be caused by the slight difference in surface tension for high concentrated acids [38,39,40] in comparison to water (ca. 73 mN/m at 20 °C). After the initial compression, the yarn diameter again increases nearly to the value of pristine yarn. It is thought that the acids slowly infiltrate the inner yarn structure with treatment time and widen the spaces between CNT bundles. However, as these bundles are difficult to infiltrate, they remain compressed even over long time, which would explain the remaining higher conductivity for half-concentrated HNO_3_, H_2_O_2_ and HCl compared to pristine yarn.

The temperature-dependent conductivity measurements after acid treatment (Figure 10) support the theory of yarn compression during acid treatment. There is no change of the slope of the curves after 3 h and 216 h for half-conc. HNO_3_, H_2_O_2_ and HCl. Only concentrated HNO_3_ shows a slight influence on the conduction mechanism, presumably due to its oxidative nature. After 216 h an increase of the slope indicates the decomposition of the CNT.

## 3. Materials and Methods

CNY were produced by a two-step dry spinning process from a multiwalled CNT (MWCNT) array. The MWCNT array were grown by chemical vapor deposition as described by Iijima et al. [23]. The parameters for spinning the yarn were 1000 turns per min with a spinning speed of 40 mm/min. The resulting yarns were determined by thermogravimetric analysis TGA and energy-dispersive X-ray spectroscopy (EDX) measurements to be free of impurities including catalyst particles. We could not verify any further elements.

Annealing of CNY was conducted in a high temperature furnace (LHTG 100-200/30-1G, GERO, Neuhausen, Germany) under an Argon atmosphere and normal pressure. For this treatment, CNY were put into closed graphite vessels. Sections of the yarns were annealed at 1000, 1500, 2000 and 2500 °C for 30, 60 and 120 min.

For functionalization, a variety of different acids were used including concentrated (65-% or 14.4 mol/L; VWR AnalaR Normapur, Dresden, Germany) and half concentrated (32-% or 6.05 mol/L; VWR AnalaR Normapur, Dresden, Germany) nitric acid, conc. hydrochloric acid (37-% or 12.02 mol/L; VWR AnalaR Normapur, Dresden, Germany) and conc. hydrogen peroxide (30-% or 9.8 mol/L; Sigma Aldrich, Darmstadt, Germany). Pieces of 3 cm long CNY were put into glass vials with an excess amount of acid (5–10 mL). The acid treatment was carried out at 25 °C for 3 to 216 h for each of the acid treatment processes. Directly after each treatment, the samples were washed with water and dried over night at 108 °C.

CNY were characterized by SEM (Nova NanoSEM 200, FEI) and TEM (TITAN, FEI) before and after each annealing and acid treatment. SEM and TEM investigations were conducted using a cathode voltage of 15 kV and 80 kV respectively. For TEM investigations we spread the CNY in a mechanical way using tweezers to individualize several CNT. These CNY samples were fixed on a copper grid with special TEM glue. We investigated serval parts of the samples on the basis of more than 20 SEM images and about 75 TEM images. For statistical analysis of the CNT diameter a well as the number of walls we evaluated were min. 100 CNT per sample (Appendix A).

Characterization with a Micro-Raman Spectrometer (Horiba Jobin Yvon, France) was performed over between 1000 and 1800 cm^−1^ using a wavelength of 514.5 nm (Argon-Laser, Coherent, Santa Clara, CA 95054, USA). I_D_/I_G_ ratios were calculated from Raman spectra by dividing the intensity of the D-band through the intensity of the G-band. For each sample, 5 Raman measurements were conducted. Electrical conductivity was measured between 5 and 295 K with a Nanovoltmeter (Keithley Instruments, Solon OH44139, USA) using the four-point measurement method. Here the current will be operated by two contacts at the ends of the CNY whereas the voltage will be measured by two additional contacts, which are located between the two current supply contacts (Appendix A). Specific electrical conductivity (*σ*) was calculated by the equation $\sigma =\frac{l}{RA}$ where *l* is the length between the inner contacts of the four-point set up, *A* is the cross-sectional area of the CNY calculated from the diameter measured from SEM images and *R* is the measured resistance.

## 4. Conclusions

In this work, we show a systematic study of the influence of annealing and functionalization by treatment with highly concentrated acids on the electrical conductivity and structure of CNY. Annealing enhances electrical conductivity by a factor of more than 5.5. A high graphitization of the CNT at 2500 °C leads to enhanced transport of electrons through the individual CNT. It was found the amorphous carbon resulting from the synthesis of the CNT plays an important role by connecting the CNT in the yarn structure and helps to reduce the contact resistance between adjacent CNT. Acid treatment over longer times and with different kinds of acids leads to an increase of electrical conductivity with a treatment time of 3 h resulting in the optimal increase. With this method, an increase of more than two times can be achieved. Furthermore, it was found that this increase is less dependent on forming functional groups on the surface of the CNT, but on the compression of the yarns and reduction of the intertube space. These results for annealing and functionalization help to understand the influences on CNT yarns by different post synthetic treatments and reveal key factors that could assist in producing highly conductive CNT yarns.

## Figures and Tables

**Figure 1 molecules-25-01144-f001:**
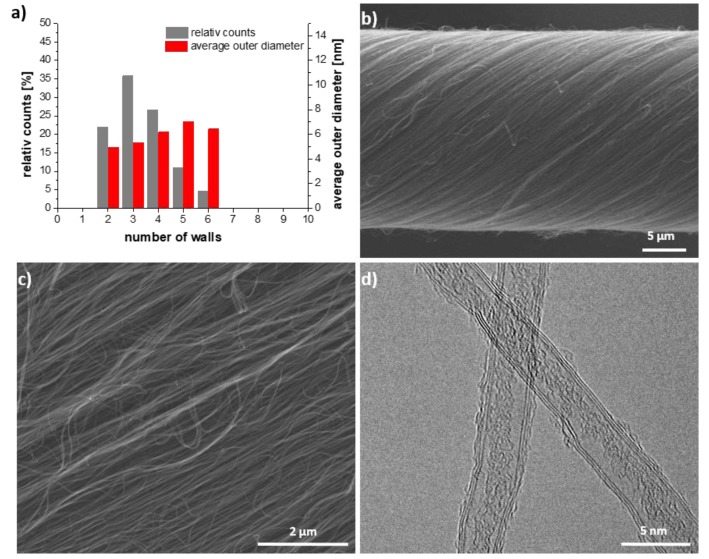
(**a**) Statistical analysis of the pristine carbon nanotube (CNT), (**b**,**c**) SEM images of pristine carbon nanotube yarn (CNY) and (**d**) TEM image of pristine CNT.

**Figure 2 molecules-25-01144-f002:**
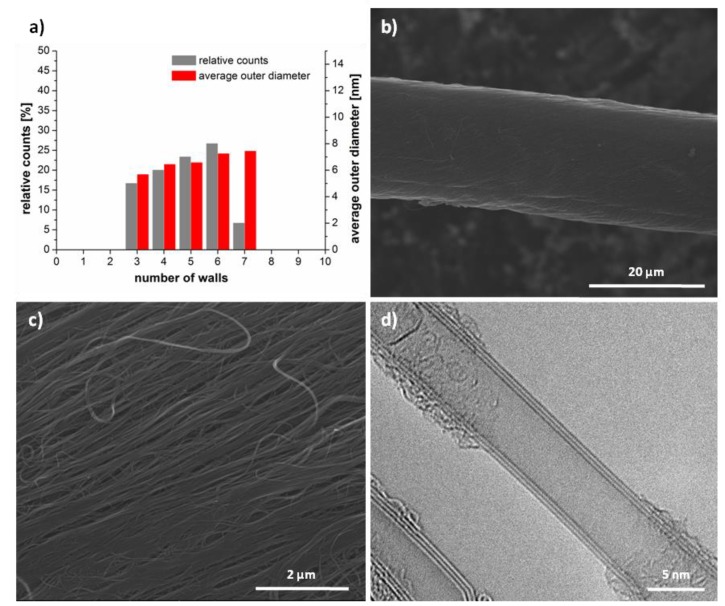
Investigation of CNY annealed for 2 h at 2500 °C. (**a**) Statistical analysis of the annealed CNT, (**b**,**c**) SEM images of annealed CNY and (**d**) TEM image of annealed CNT.

**Figure 3 molecules-25-01144-f003:**
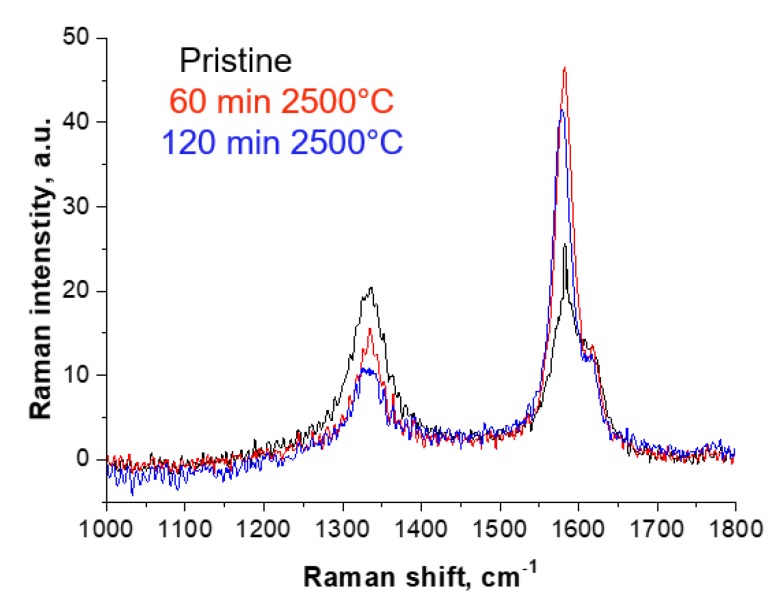
Raman spectra of pristine CNY and of annealed CNY for 60 and 120 min at 2500 °C.

**Figure 4 molecules-25-01144-f004:**
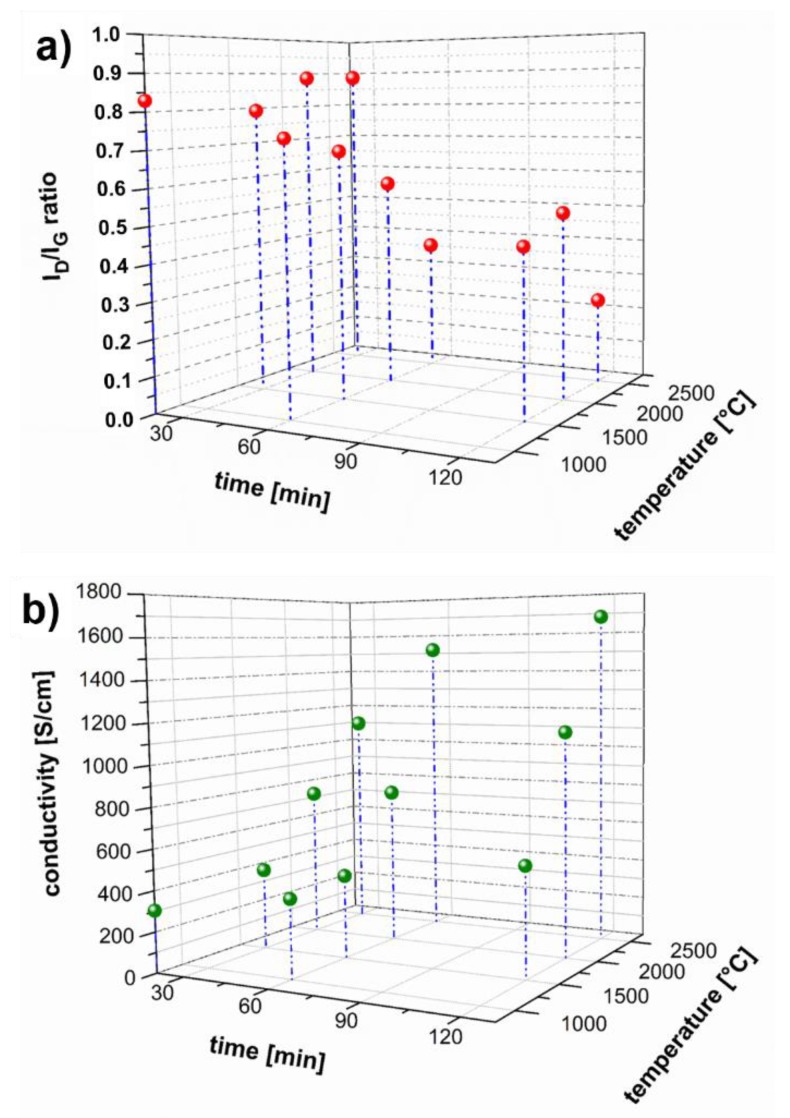
(**a**) Raman spectroscopy I_D_/I_G_ ratio as a function of annealing time and temperature, (**b**) electrical conductivity of CNT yarns as function of annealing time and temperature.

**Figure 5 molecules-25-01144-f005:**
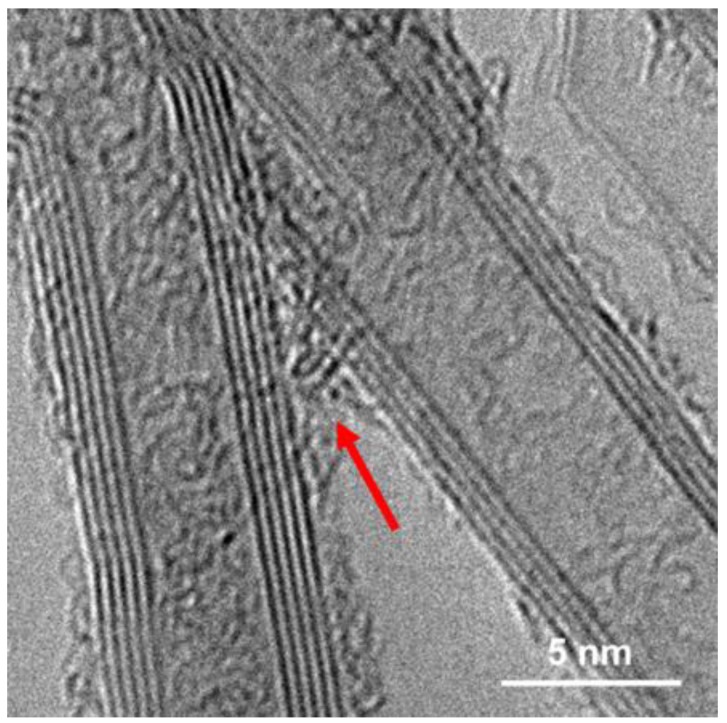
Bridging of adjacent CNT by amorphous carbon after annealing at 2500 °C for 2 h.

**Figure 6 molecules-25-01144-f006:**
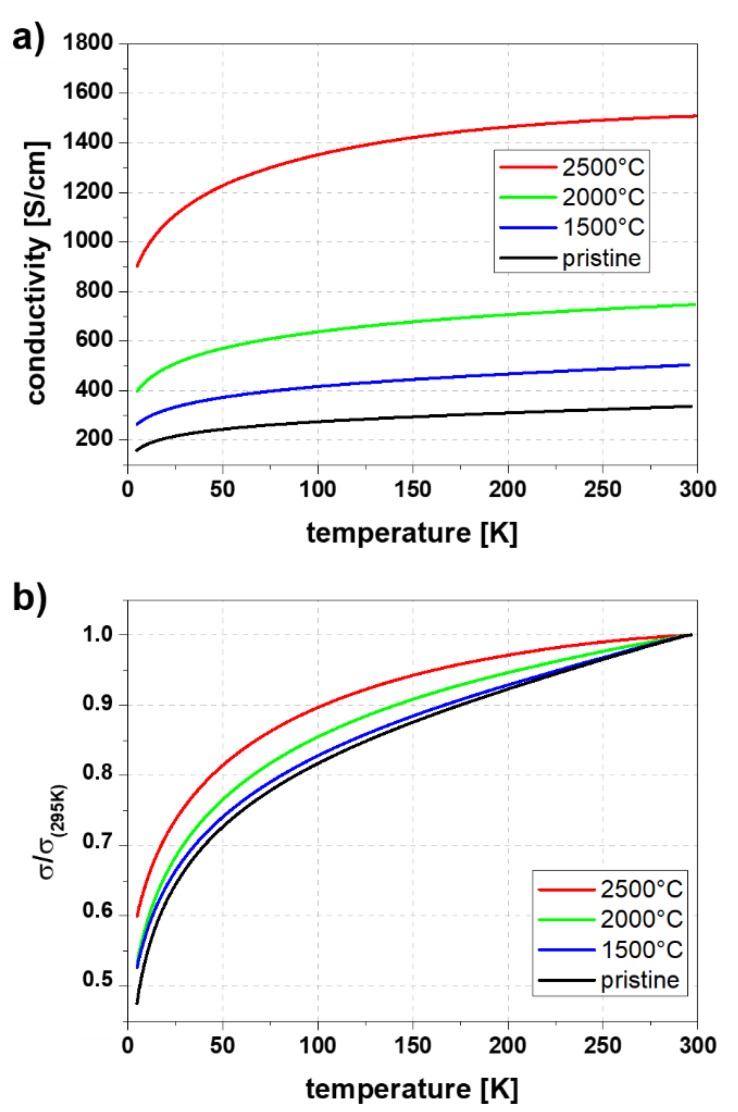
(**a**) Specific and (**b**) normalized electrical conductivity of pristine and 2 h annealed CNT yarns.

**Figure 7 molecules-25-01144-f007:**
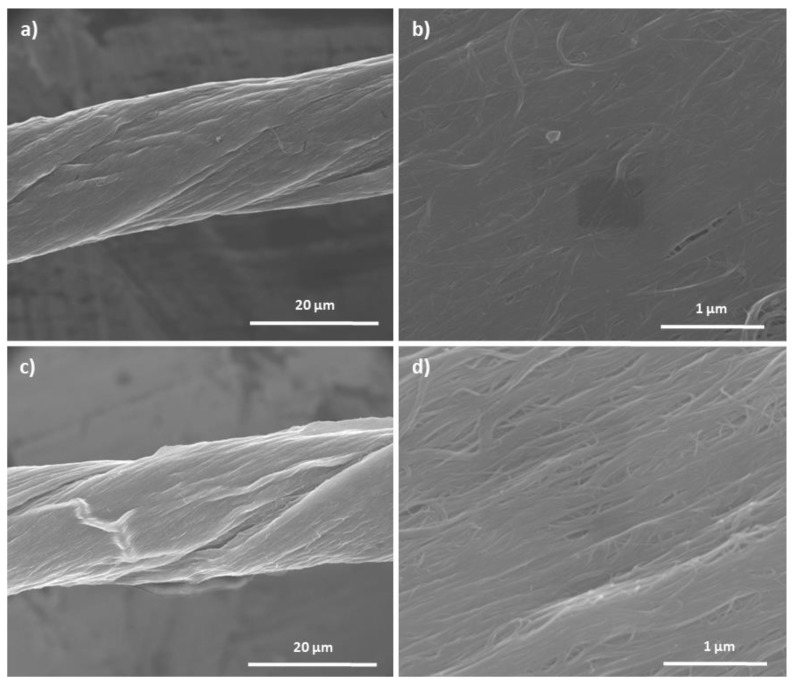
SEM images of yarns after treatment with half concentrated HNO_3_: (**a**,**b**) after 3 h and (**c**,**d**) after 216 h treatment time.

**Figure 8 molecules-25-01144-f008:**
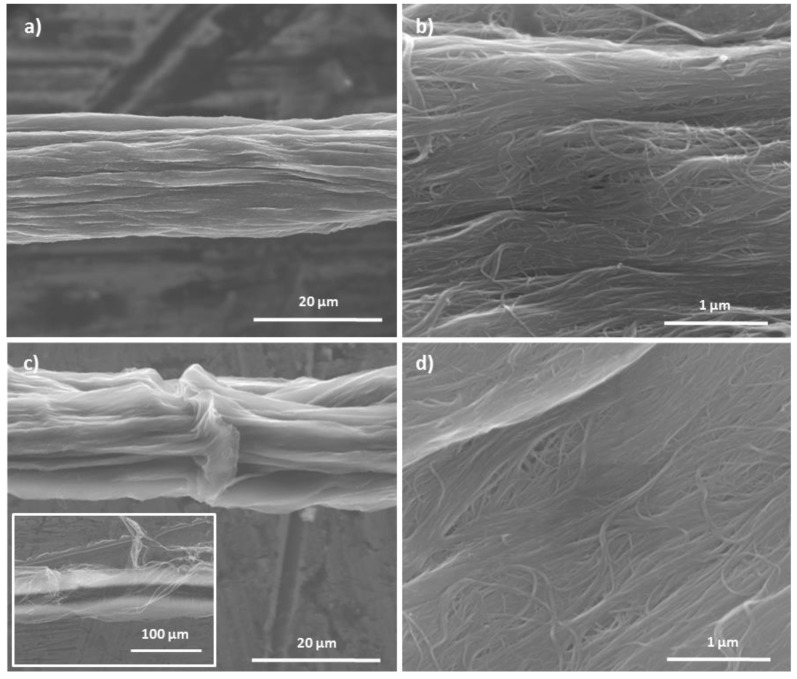
SEM images of yarns after treatment with concentrated HNO_3_: (**a**,**b**) after 3 h and (**c**,**d**) after 216 h treatment time.

**Figure 9 molecules-25-01144-f009:**
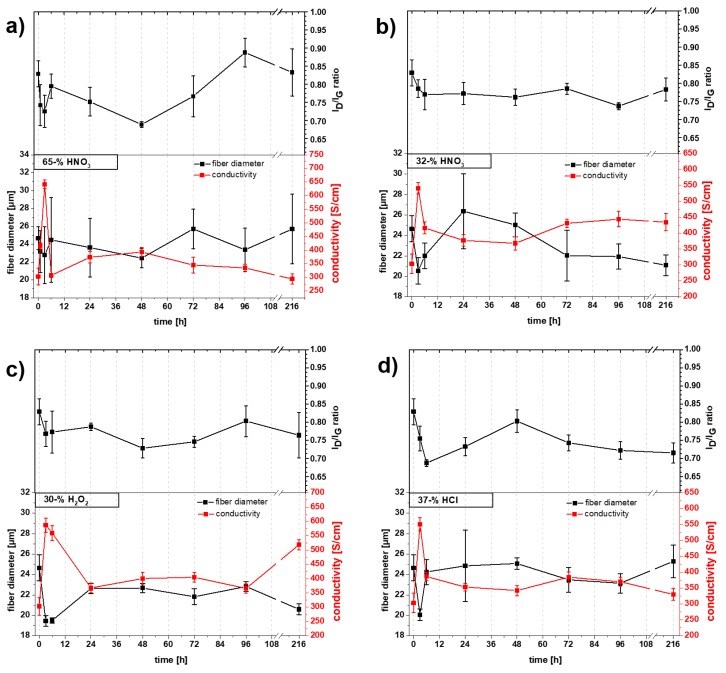
Conductivity, I_D_/I_G_ ratio and fiber diameter of CNY after different acidification times: (**a**) concentrated nitric acid, (**b**) half concentrated nitric acid, (**c**) concentrated hydrogen peroxide and (**d**) concentrated hydrochloric acid.

**Figure 10 molecules-25-01144-f010:**
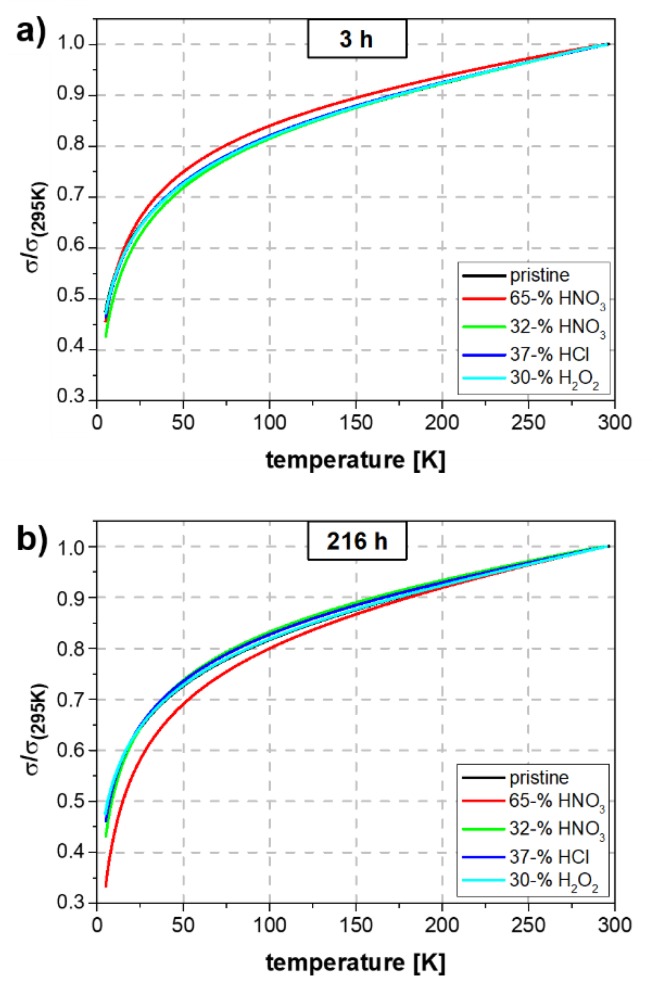
Normalized electrical conductivity of pristine yarn and yarns after 3 (**a**) and 216 h (**b**) of acid treatment.

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
