# Peer review of "Systematic Investigations of Annealing and Functionalization of Carbon Nanotube Yarns"

_molecules, 2020, doi:10.3390/molecules25051144_

Round 1

Reviewer 1 Report

In the paper entitled “Systematic investigations of annealing and functionalization of carbon nanotube yarns”, authors provided an interesting study on CNYs and how thermal annealing and acid purification affect their electrical properties. Some concerns should be addressed before its publication.

Decision: Major Revision

Why thermal treatment for 30 minutes increases ID/IG @ 2000 and 2500C? Is it some experimental error or it can be justified (it is also observed for 2h when temperature rises from 1500C to 2000C.
In Figure 4, authors claim that amorphous carbon bridging can enhance the electrical conductivity of their samples. Amorphous carbon usually refers to carbon with sp3 hybridization and it is believed to deteriorate electrical conductivity. How bridging two conductive CNT with non-conductive amorphous carbon can increase conductivity.
Authors should provide some samples of their Raman spectra to clarify their procedure and make a comparison between peak positions etc.
The authors claimed that their CNTs are free of catalyst particles but as it is described in the original paper, iron catalyst particles do exist in CNT roots. The authors can validate their claim by performing a simple TGA test under air atmosphere (900-950C) and evaluate the remaining mass as the catalyst percentage in their sample.
The authors should provide more information on how they measured the electrical conductivity of their samples. CNT samples are fluffy and very difficult to handle. Do they prepare rigid samples by compression? Because measuring electrical conductivity with 4- probe on fluffy CNT samples is practically very difficult and erroneous.
If the authors tried purification to increase electrical conductivity, why they did not do the purification stage prior to the thermal treatment? Because the treatment of CNY with acids increases the chance of defects formation and functionalization. If they performed acid treatment before thermal annealing, there would be a higher chance to obtain higher electrical conductivity. The authors should try at least for one conditions and see if this improve their results or not.
To eliminate amorphous carbon in their samples, I suggest to do thermal annealing at lower temperatures (<400C) in air atmosphere according to their TGA data for their samples. I think after this stage and doing thermal annealing in an inert atmosphere they could obtain higher electrical conductivities.
Please edit the references in line 72 and 74.

Please correct line the definition of Specific electrical conductivity in line 286

Reviewer 2 Report

In this study, the authors present a systematic study of the effects of annealing and functionalization by treatment with highly concentrated acids on the electrical conductivity and structure of CNY. Representative methods of improving the electrical conductivity of CNY include heat treatment and functionalization. In this study, the experimental results are presented for this representative method. However, in order to understand the treatment method more carefully, the results of Raman are required for structural analysis before and after heat treatment. In addition, the results of elemental analysis are required before and after the functionalization treatment. After that, the comparative analysis of the results and the results of SEM, etc. seems to be easier to understand. This needs to be supplemented.

Reviewer 3 Report

The work described in this article has been studied in the past to various degrees of detail, as mentioned by the authors. This particular manuscript aims to add on to the body of work via a fairly thorough Raman spectra analysis of the CNY as a result of acid-functionalization and/or thermal annealing, but there needs more work to be done to expand on the findings. Some specific comments are below:

1) The English used in the manuscript and supplementary material needs to be improved, especially when it comes to grammar, the use of symbols (C vs. °C, for example), and use of tenses and punctuation.

2) Some references are missing, as indicated in lines 72-74. There are also several relevant references from groups including the Pasquali group at Rice University, Barron group at Swansea University, Page group at the University of Newcastle, Tehrani group at the University of New Mexico etc.

3) More information on the CNT synthesis and CNY spinning is beneficial to this paper. I appreciate that the authors are citing their own work from before instead, but given how important it is to this work, some of that needs to be included here for context to the reader.

4) The authors claim the CNY is completely free of catalyst particles, which is not supported by any characterization and is a bold claim to make thus. This work will need elemental analysis and thermogravimetric analysis at various stages to support this claim, and also others throughout the course of the text.

5) What was the operating pressure for the annealing processes?

6) What is the metric of acid concentration? A percentage value means nothing without more context. What is the source of the chemicals? This information is critical to those wanting to replicate this work in the future.

7) What is "room temperature" for the acid treatment experiments? This can vary from lab to lab, and actual numbers are always preferable here.

8) What was the operating voltage for SEM characterization? Was there a particular type of TEM grid/holder used for the CNY samples, and was there any sample treatment done to get a thin section of the CNTs in the TEM?

9) Given the CNTs in the CNY are being reported to have between 2 to 6 walls, using just 514.5 nm excitation for Raman spectroscopy is not going to be representative of most of the CNTs in the yarn. Higher laser wavelengths will help with resonance, and ensure that the Raman data is more typical of the entire yarn.

10) Line 286 appears to have a translation error with the symbol for specific conductivity not appearing correctly.

11) Diameter measurement from SEM is not the best way to do this, especially for specific conductivity, but adding in a mean and standard deviation for the CNY diameter will help the author's case in showing whether the yarn was consistent in diameter across the length of the samples. For example, the authors claim the CNY diameter was 25 micrometers on average, but figure 1b clearly shows the yarn has a larger diameter than that.

12) How many SEM and TEM measurements were done for each sample? The increase in number of walls for the CNTs in the yarn post annealing has to be backed by more TEM data, since the two provided show the same number of walls for the pristine CNTs and the annealed ones.

13) If figures 1b and 2b are accurate, there is a tremendous decrease in yarn diameter post thermal annealing, and far more than the 1 micron claimed.

14) It would be good to see typical Raman spectra instead of just the Id/Ig charts, perhaps as supplementary material?

15) I agree with the author's assessment of a typical 3D variable electron hopping taking place in this system, but the aforementioned questions on diameter distribution cause some doubt on the specific conductivity changes. I urge the authors to clarify these issues to remove these queries.

16) Have the authors studied the conductance of the yarns as the temperature cools from a higher temperature to a lower one as well?

17) In addition to the pristine and annealed yarn, elemental analysis and TGA is all the more necessary with the acid-treated yarns, if only to show whether there is functionalization but also ad/absorption of the acid functional groups.

Round 2

Reviewer 1 Report

The authors clarify my concerns about their research. So I accept the paper for publication in Molecules journal.

Reviewer 3 Report

Thank you for attempting to make the changes requested. I am satisfied with some of them, but unfortunately the two main issues I have are still not resolved:

1) In the absence of more resonant Raman spectroscopy using different excitation wavelengths, there is not enough evidence to support your claims on the relationship between the various annealing/acid-functionalization processes and electrical conductivity as you wish to explain via Raman spectroscopy and the trend of Id/Ig.

2) There is still no data provided to show the samples were without any impurities, which can be a big factor here.
